# Modified-Chronic Disease Score (M-CDS): Predicting the individual risk of death using drug prescriptions

Marica Iommi[1]*, Simona Rosa[1], Michele Fusaroli[2], Paola Rucci[1], Maria Pia Fantini[1], Elisabetta Poluzzi[2]

1 Department of Biomedical and Neuromotor Sciences–Hygiene and Biostatistics Unit, University of Bologna, Bologna, Italy, 2 Department of Medical and Surgical Sciences–Pharmacology Unit, University of Bologna, Bologna, Italy

* marica.iommi2@unibo.it

## Abstract

### Background

Estimating the morbidity of a population is strategic for health systems to improve health-care. In recent years administrative databases have been increasingly used to predict health outcomes. In 1992, Von Korff proposed a Chronic Disease Score (CDS) to predict 1-year mortality by only using drug prescription data. Because pharmacotherapy underwent many changes over the last 3 decades, the original version of the CDS has limitations. The aim of this paper is to report on the development of the modified version of the CDS.

### Methods

The modified CDS (M-CDS) was developed using 33 variables (from drug prescriptions within two-year before 01/01/2018) to predict one-year mortality in Bologna residents aged ≥50 years. The population was split into training and testing sets for internal validation. Score weights were estimated in the training set using Cox regression model with LASSO procedure for variables selection. The external validation was carried out on the Imola population. The predictive ability of M-CDS was assessed using ROC analysis and compared with that of the Charlson Comorbidity Index (CCI), that is based on hospital data only, and of the Multisource Comorbidity Score (MCS), which uses hospital and pharmaceutical data.

### Results

The predictive ability of M-CDS was similar in the training and testing sets (AUC 95% CI: training [0.760–0.770] vs. testing [0.750–0.772]) and in the external population (Imola AUC 95% CI [0.756–0.781]). M-CDS was significantly better than CCI (M-CDS AUC = 0.761, 95% CI [0.750–0.772] vs. CCI-AUC = 0.696, 95% CI [0.681–0.711]). No significant difference was found between M-CDS and MCS (MCS AUC = 0.762, 95% CI [0.749–0.775]).

emilia-romagna.it/) and, although they are anonymized, datasets are not publicly available due to the current regulation on privacy. The description of the administrative databases is available from the website https://salute.regione.emilia-romagna.it/siseps/sanita/asa/documentazione. Other researchers can obtain access to the data through a formal request based on a research project to the Emilia-Romagna Regional Health Agency. We otbtained the access to the data in the framework of a research agreement between the University of Bologna and the Emilia-Romagna Regional Health Agency entitled "Economic assessment of the diagnostic and therapeutic pathways to care (PDTA) and appropriateness of drug prescriptions". The database includes the following variables: patient ID, age, gender, area of residence, 33 dummy variables for drug prescriptions, death at one year (yes/no) and days of follow-up.

**Funding:** The authors received no specific funding for this work.

**Competing interests:** The authors have declared that no competing interests exist.

## Conclusions

M-CDS, using only drug prescriptions, has a better performance than the CCI score in predicting 1-year mortality, and is not inferior to the multisource comorbidity score. M-CDS can be used for population risk stratification, for risk-adjustment in association studies and to predict the individual risk of death.

## Introduction

Estimating the morbidity status of a population is crucial for public health, in order to manage people with multiple chronic diseases efficiently and effectively. Multimorbidity is defined as the presence of multiple (chronic or acute) diseases and medical conditions in one individual [1].

In recent years there has been an increasing use of administrative databases as data sources for conducting clinical and pharmaco-epidemiological studies [2]. The advantages of administrative databases include their immediacy to be analysed, the good reliability, the wide geographical coverage, the long-term follow-up and the good detail of the clinical history of the individual [3].

Administrative databases, however, have some limitations, including the lack of information on the lifestyle, the social and economic characteristics and the presence of bias related to their observational nature.

In the past, the most popular comorbidity indices, i.e. the Charlson Comorbidity Index–CCI [4] and the Elixhauser Index–EI [5] were developed using the diagnoses reported in the hospital discharge records, coded through the International Classification of Diseases (ICD) system.

In 1992, von Korff et al. [6] developed a drug-based index, the Chronic Disease Score (CDS), to predict health outcomes. It originally consisted of 17 diseases with a weighing system assigned a priori and was subsequently updated by Clark et al. [7] to include 28 categories with a weighing system based on regression models. No further updates have been made, inevitably leading to limitations in the use of the score, as new drugs have been introduced in the market to treat chronic diseases (e.g., monoclonal antibodies for autoimmune disorders). Furthermore, in drug prescription databases specific criteria (e.g., amount threshold) to select chronic patients are required.

Several studies have compared the predictive ability of diagnosis-based indexes to drug-based ones, however, the superiority of an approach in predicting health status has not been demonstrated yet [8, 9], and the lack of an updated version of CDS does not allow appropriate performance assessment and comparisons [10].

Corrao et al. [11] proposed the Multisource Comorbidity Score (MCS), that combines information from hospital discharge records and pharmaceutical prescriptions to stratify individuals according to their morbidity profile. The MCS proved to be a better predictor of 1-year mortality compared to the CI, EI and CDS indices.

Merging different databases on an individual level is a very demanding computational work. Using only pharmaceutical database may decrease the computational workload while capturing the complexity of patients' clinical condition.

The main aim of this study was to implement a new version of the Chronic Disease Score, using detailed information from the pharmaceutical prescription databases that incorporates, in addition to traditional drug treatments, the novel pharmacotherapies introduced over the last 3 decades and the amount of drugs consumed by the individual.

## Methods

### Data source

The population was extracted from the administrative databases of Emilia-Romagna Regional Health Agency (https://salute.regione.emilia-romagna.it/siseps/sanita), a Northern Italian region with approximately 4.4 million Italian citizens, who have universal access to the Italian National Health System. The retrospective study was carried out in conformity with the regulations on data management with the Italian law on privacy (Legislation Decree 196/2003 amended by Legislation Decree 101/2018). Data were anonymized prior to the analysis at the regional statistical office, where each patient is assigned a unique identifier. This identifier does not allow others to trace the patient's identity and other sensitive data. Anonymized regional administrative data can be used without a specific written informed consent when patient information is collected for healthcare management and healthcare quality evaluation and improvement (according to art. 110 on medical and biomedical and epidemiological research, Legislation Decree 101/2018). The study was approved by the Independent ethics committee of the Larger Area Emilia Center (CE-AVEC) of the Emilia-Romagna Region, study protocol 398/2019/Oss/AOUBo on 19/06/2019.

Data were retrieved from the Regional Health Authority Outpatient Specialty Database (OSD) of 2016–2017, which includes all specialty visit and laboratory tests. Inclusion criteria were: (1) residence in the catchment area of the Local Healthcare Authorities (LHA) of Bologna or Imola ($\approx$1,013,000 inhabitants) as of 1st January 2018, (2) age $\geq$50 years, (3) having received at least one outpatient service in the previous 18 months. Patients were followed-up from 01/01/2018 to 31/12/2018 and censored at death or at the end of follow-up. The date of death was retrieved from the Regional Mortality Registry Database (MR) of 2018.

Using a unique pseudonymized patient code, the demographic characteristics of the study cohort identified through the OSD database were linked with the Regional Outpatient Pharmaceutical Database (OPD) of 2016–2017, which includes the drugs reimbursed by the NHS (prescribed by the primary care physician or a specialist, or directly dispensed by the hospital pharmacies) and details on substance name, Anatomical Therapeutic Chemical (ATC) classification system code-V.2013, brand name, date of prescription filling, number of unit doses and number of packages and prescribers.

To compute the Charlson Comorbidity Index and the Multisource Comorbidity Score, the Hospital Discharge Record (HDR) database of 2016–2017 were also linked with the OSD, which contains information on admission and discharge dates, diagnosis and interventions (identified using the International Classification of Diseases, 9th revision, Clinical Modification—ICD-9-CM coding system), and discharge status.

### Algorithm and score development

The modified CDS (M-CDS) was developed by updating the Chronic Disease Score proposed by Von Korff et al. [6] with the currently marketed drugs (up to March 2019; https://farmaci.agenziafarmaco.gov.it/bancadatifarmaci/home, of the Italian Medicines Agency) and the list of drug classes refunded by NHS, according to the updated therapeutic guidelines for chronic diseases.

We referred to three key-studies that use pharmacy data to estimate the prevalence of chronic conditions [11–13]. After reviewing these studies, the following decisions were made:

1. Conditions that could not be discerned on the basis of pharmacotherapy were merged (i.e. Alzheimer disease was merged to dementia, Crohn's disease to inflammatory bowel diseases, hypertension to Cerebrovascular disease);

2. Drugs prescribed for conditions different from the ATC target class were removed (i.e. anti-tussives were removed from cancer) or assigned to other conditions (i.e. benzodiazepines were moved from Parkinson to a condition grouping anxiety, depression and obsessive compulsive disorders, apart from midazolam which was moved to epilepsy; rifabutin was moved from HIV to tuberculosis; antidepressants were moved from psychosis to depression);

3. Drug not listed in the above-mentioned studies, such as liver therapy in liver diseases, or antivirals for HCV in chronic hepatitis, were added after a thorough in-depth review of drug classes refunded by NHS;

4. Conditions that could be unambiguously associated with one or more drugs, such as multiple sclerosis, were also added to the list.

Using these criteria, 60 conditions unambiguously associated to pharmaceutical prescriptions were identified; 33 were selected on an epidemiologic rationale based on incidence (e.g. acromegaly was excluded) and on survival expectancy in the older age (e.g. spinal muscular atrophy was excluded). Cystic fibrosis was retained to be consistent with the MCS score (i.e. Corrao *et al.*, 2017), although a low incidence is expected in adults. We decided to distinguish between generic prescriptions for exocrine pancreas failure, that can be related to multiple aetiology, and prescriptions specific to cystic fibrosis.

To limit the inclusion of occasional drug users, a cut-off on the minimum number of prescriptions was set for the years 2016 and 2017, depending on the drug. The list of candidate condition, the corresponding ATC codes and the minimum number of prescriptions are reported in S1 Table. Thresholds of number of prescriptions are defined on the basis of recommended dosing schemes, packages available on the market and relevant prescribing/reimbursing rules.

In order to estimate the weights for the 33 conditions included in the M-CDS, the study population resident in Bologna was randomly split into a training set (80%) and a test set (20%). A Cox regression model to predict one-year mortality was developed in the training set. The LASSO (Least Absolute Shrinkage and Selection Operator) method was applied to select only relevant covariates among gender, age and the 35 conditions recorded in the 2 previous years (2016–2017) [14]. The best algorithm was achieved through 10-fold cross-validation, that chooses the algorithm with the smallest cross-validated mean squared error [15]. To reduce the risk of overfitting, we selected the most parsimonious model, that is within one standard error of the best model [13].

The comorbidity weights were obtained by multiplying the regression coefficients of the Cox model by 10 and rounding them off to the nearest integer number. The M-CDS total score was computed as the sum of the comorbidity weights.

Finally, a classification tree analysis with chi-square automatic interaction detection (CHAID) growing process (maximum tree depth = 3; minimum cases in parent node = 1000, child node = 500; significance level of spitting and merge set to 0.05) was used to identify optimal cut-points of the total M-CDS to stratify the population according to the risk of death [16].

## Validation of the M-CDS

The M-CDS was internally validated on the test set sample of Bologna population and then was externally validated in the population of Imola. The log-rank test was used to compare the survival distributions of patients in the M-CDS classes.

The performance of the M-CDS and its discriminant ability were compared between the test set (residents of Bologna) and those of the CCI and MCS using the c-statistic. As a secondary outcome, the ability of M-CDS to predict 1-year hospitalization was evaluated.

The HDR database of 2016 and 2017 was used to reproduce the weighted score of CCI and of MCS (combining HDR with the pharmaceutical databases) to predict mortality in 2018.

The significance level for all the analyses was set at p<0.05. Statistical analyses were performed using R, version 3.6.3 and IBM SPSS version 25.0.

## Results

### Characteristics of the study population

We identified 436,561 individuals aged ≥ 50 years, resident in the LHAs of Bologna and Imola, alive on January 1st, 2018, with at least 1 outpatient service access in the previous 18 months. The final study population therefore includes 380,849 residents in the LHA of Bologna and 55,712 residents in the LHA of Imola (≈94.4% of Bologna and Imola total residents). Table 1 provides a description of the two LHA study populations.

Patients had a mean age of 67.5 years in the LHA of Bologna and 67.1 in the LHA of Imola, in both LHA there was a predominance of females (56.1%-55.3%). During 2018 around 2.4% and 2.3% individuals died in the LHA of Bologna and Imola, respectively.

The most frequent conditions were Cerebrovascular disease (57.1%-59.9%), hyperlipidaemia (25–26.4%), acid related disorders/peptic ulcer (24.8%-28.6%) and pain and inflammation (21.2%-24.5%). Cystic fibrosis and additive disorders were excluded from the model as candidate predictors since only one case was found in the LHA of Bologna and zero cases in LHA of Imola. Cases that other scores would have put in cystic fibrosis have been relocated to the more generic label exocrine pancreas failure.

### M-CDS total score weights

In Table 2, coefficients and weights of the multiple Cox regression model are presented. The LASSO method selected 18 variables over the 31 conditions (cystic fibrosis and addictive disorders were excluded because they were virtually absent in the population). The variables that mostly contributed to the total score were cancer, chronic renal disease and psychosis. Acid related disorders/peptic ulcer, respiratory illness, exocrine pancreas failure and liver diseases were significant predictors of mortality but with small contributions to the total score. In the training set, the mean M-CDS score was 3.4±4.1 (median = 2; IQR [0–5]; range [0–45]).

The classification tree analysis with CHAID growing process suggested to split the M-CDS score into 6 mutually exclusive classes (≤1, 2, 3–4, 5–6, 7–9 and ≥10). In the training population of Bologna, 9.3% individuals had a M-CDS value ≥10 and 35.6% a value ≤1 (Fig 1).

The distribution of M-CDS was similar between males and females in each age group and became flatter in the older age groups, starting from 70 years (Fig 2).

### Predictive performance

The ability of M-CDS to predict 1-year mortality was similar between the Bologna training set (AUC 0.765; 95% CI [0.760–0.770]) and the test set (AUC 0.761; 95% CI [0.750–0.772]), as well as in the Imola population (AUC 0.768; 95% CI [0.756–0.781]), confirming the stability of the score. Fig 3 shows the ROC curves of M-CDS in the three samples.

The 1-year survival in the test set differed significantly among the M-CDS classes (log-rank test chi-square = 2106.8; p<0.001), and decreased with the increase of M-CDS class from 99.5% in class with score ≤1 to 91.1% in class with score ≥10 (Fig 4).

Fig 5 shows the ROC curves for M-CDS, MCS and CCI predicting 1-year mortality, in the test set. The M-CDS (AUC 0.761; 95% CI [0.750–0.772]) was superior to the CCI (AUC 0.696;

**Table 1. Demographic and characteristics of the Bologna and Imola residents aged ≥50 years.**

| | LHA of Bologna (n = 380,849) | LHA of Imola (n = 55,712) |
|---|---|---|
| **Age (mean ± S.D.)** | 67.5±11.8 | 67.1±11.7 |
| **Females (n, %)** | 213,572 (56.1%) | 30,811 (55.3%) |
| **Condition (n, %)** | | |
| Cardiovascular and cerebrovascular disease | 217,590 (57.1%) | 33,372 (59.9%) |
| Respiratory illness | 49,867 (13.1%) | 7,982 (14.3%) |
| Exocrine pancreas failure | 1,170 (0.3%) | 212 (0.4%) |
| Cystic fibrosis | 1 (0%) | 0 (0%) |
| Tuberculosis | 349 (0.1%) | 44 (0.1%) |
| Cancer | 13,718 (3.6%) | 1,841 (3.3%) |
| Acid related disorders/peptic ulcer | 94,458 (24.8%) | 15,940 (28.6%) |
| Irritable colon | 1,515 (0.4%) | 234 (0.4%) |
| Liver diseases | 8,227 (2.2%) | 718 (1.3%) |
| Chronic hepatitis | 689 (0.2%) | 106 (0.2%) |
| Diabetes | 37,256 (9.8%) | 5,830 (10.5%) |
| Glaucoma | 25,358 (6.7%) | 3,937 (7.1%) |
| Chronic renal disease | 434 (0.1%) | 70 (0.1%) |
| Anaemias | 14,926 (3.9%) | 2,146 (3.9%) |
| Bone diseases | 26,136 (6.9%) | 5,143 (9.2%) |
| Inflammatory bowel + rheumatologic disease | 2,901 (0.8%) | 426 (0.8%) |
| Pain and inflammation | 80,737 (21.2%) | 13,648 (24.5%) |
| Hyperuricemia/gout | 25,482 (6.7%) | 4,595 (8.2%) |
| Dermatological severe | 7,170 (1.9%) | 1,159 (2.1%) |
| Transplantation | 473 (0.1%) | 65 (0.1%) |
| Hyperlipidaemia | 95,232 (25%) | 14,703 (26.4%) |
| HIV | 886 (0.2%) | 110 (0.2%) |
| Hypothyroidism | 35,344 (9.3%) | 4,997 (9.0%) |
| Epilepsy | 4,888 (1.3%) | 694 (1.2%) |
| Dementia | 2,943 (0.8%) | 236 (0.4%) |
| Parkinson's disease | 4,933 (1.3%) | 709 (1.3%) |
| Depression, anxiety, obsessive-compulsive disorder (OCD) | 56,376 (14.8%) | 7,722 (13.9%) |
| Bipolar disorders | 713 (0.2%) | 87 (0.2%) |
| Psychosis | 9,108 (2.4%) | 1,443 (2.6%) |
| Multiple sclerosis | 160 (0%) | 34 (0.1%) |
| Haemorrhagic diathesis | 1,424 (0.4%) | 300 (0.5%) |
| Allergic disorders | 13,839 (3.6%) | 2,012 (3.6%) |
| Addictive disorders | 1 (0%) | 0 (0%) |

95% CI [0.681–0.711]) ($\chi^2$ test = 88.14; p<0.001), while it did not differ from the MCS (AUC 0.762; 95% CI [0.750–0.774]) ($\chi^2$ test = 0.06; p = 0.8144).

## Secondary outcome

We also evaluated 1-year hospitalization as secondary outcome. In the Bologna training set the AUC was 0.710 (95% CI [0.660–0.760]), slightly higher than the test set (AUC 0.695; 95% CI [0.594–0.795]) and the Imola population (AUC 0.667; 95% CI [0.570–0.764]).

**Table 2. Coefficients and weights of the multiple Cox regression model to predict one-year mortality in the training population of Bologna.**

| Selected variables | Coefficient | Weight |
|---|---|---|
| Cancer | 1.039 | 10 |
| Chronic renal disease | 0.689 | 7 |
| Psychosis | 0.677 | 7 |
| Haemorrhagic diathesis | 0.446 | 4 |
| Depression, anxiety, OCD | 0.440 | 4 |
| Epilepsy | 0.422 | 4 |
| Anaemias | 0.422 | 4 |
| Parkinson's disease | 0.325 | 3 |
| Diabetes | 0.293 | 3 |
| Gout | 0.286 | 3 |
| Irritable colon | 0.282 | 3 |
| Transplantation | 0.256 | 3 |
| Dementia | 0.244 | 2 |
| Cardiovascular and cerebrovascular disease | 0.177 | 2 |
| Acid related disorders/peptic ulcer | 0.139 | 1 |
| Respiratory illness | 0.108 | 1 |
| Exocrine pancreas failure | 0.087 | 1 |
| Liver disease | 0.082 | 1 |

The M-CDS, evaluated in the test set, did not significantly differ from either the CCI (AUC 0.659; 95% CI [0.517–0.801]; $\chi^2$ test = 0.21; p = 0.647) or the MCS (AUC 0.786; 95% CI [0.673–0.899]; $\chi^2$ test = 2.14; p = 0.143).

## Discussion

We developed an updated version of the CDS that includes up-to-date outpatient drug prescriptions based on recent guidelines for chronic diseases and relevant currently marketed drugs, to predict one-year mortality.

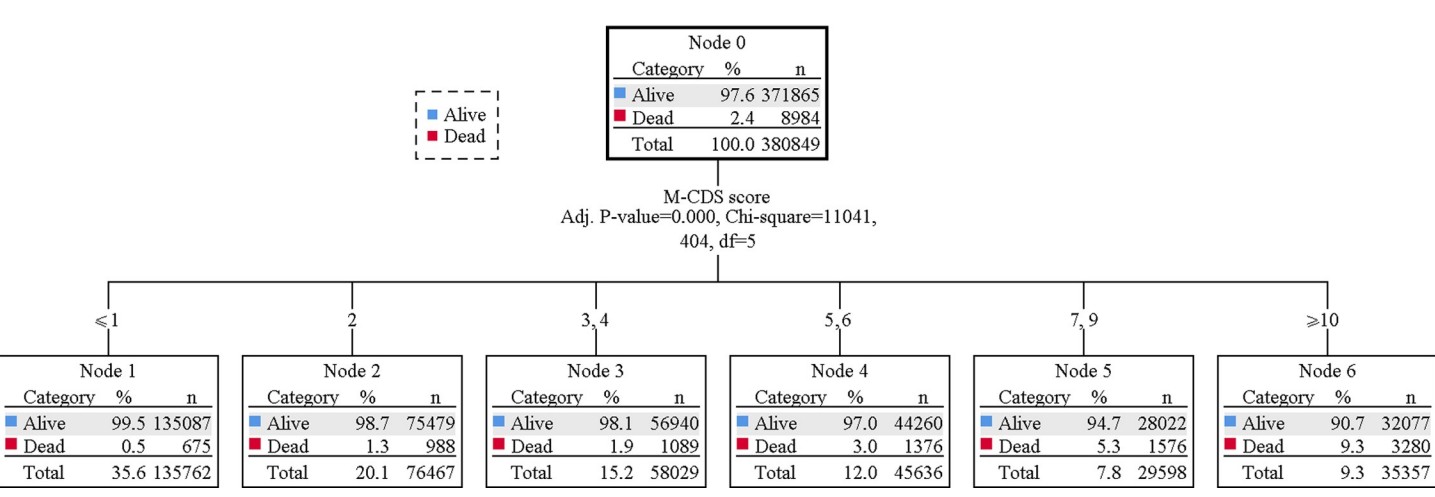

**Fig 1. Tree plot of the classification tree analysis with CHAID growing process.**

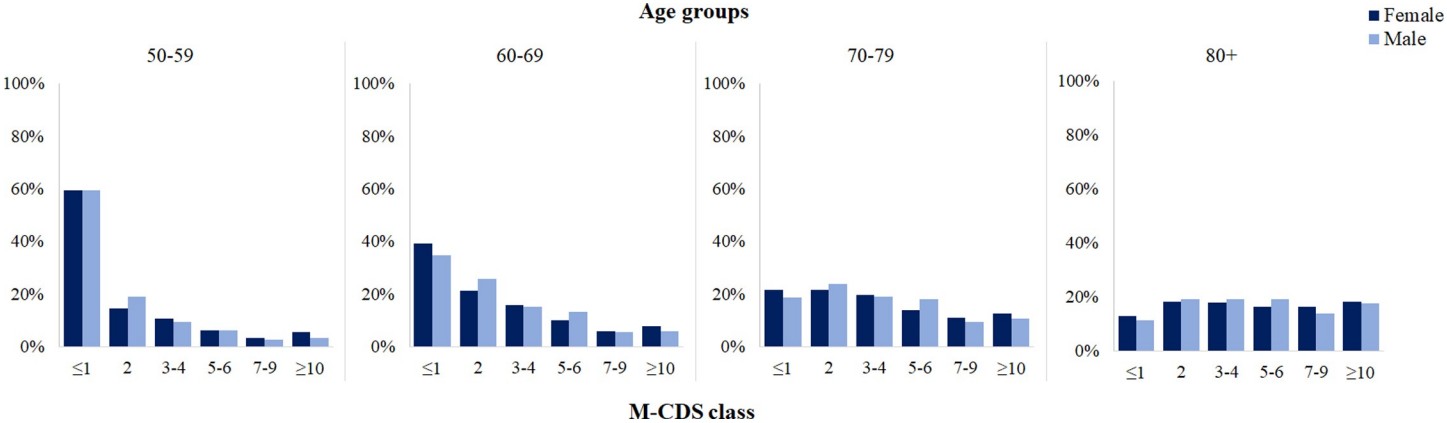

**Fig 2. Distribution of the M-CDS classes by age groups and gender.**

It was derived from the CDS first published in 1992, and its update in 1995. In this study it has been further updated by adding single drug classes (or single agents) to already listed diseases and other chronic diseases, splitting previous categories when appropriate.

The Modified-Chronic Disease Score showed a good performance in predicting individual risk of death in a very large population (individuals aged ≥50 years). Of the 33 variables investigated, corresponding to as many chronic conditions and to 80 drugs (or drug classes), 18 were selected by the analytic procedure as relevant predictors of mortality. Cancer, chronic renal disease and psychosis were the conditions that most contributed to the total M-CDS score (10 to 7 points) while acid related disorders/peptic ulcer, respiratory illness, exocrine pancreas failure and liver diseases contributed less (1 point). These 18 conditions allowed to stratify the population into 6 mutually exclusive classes, with different 1-year survival.

Our results underscore the relevant role of mental disorders as predictors of mortality, consistent with Corrao et al. [11]. A comprehensive meta-analysis of mortality related to mental disorders showed that these patients have a mortality rate 2.22 times higher than the general

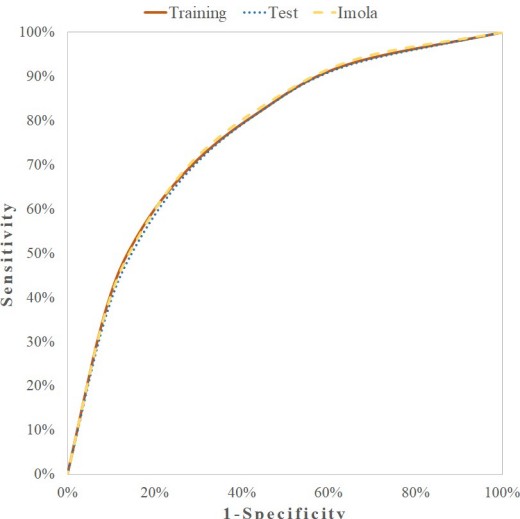

**Fig 3. ROC curves comparing the ability of M-CDS to predict 1-year mortality in the training and test sets and in Imola population.**

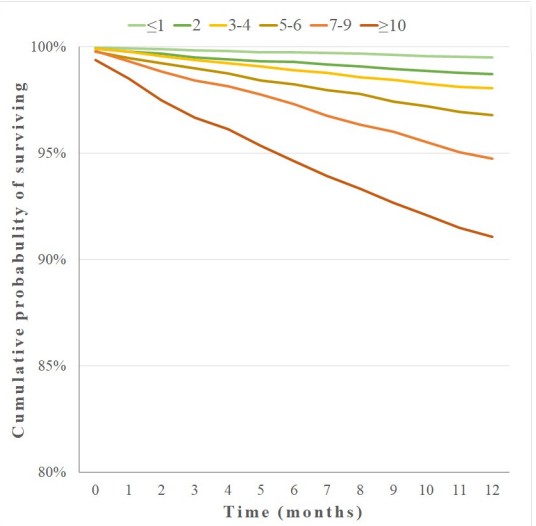

**Fig 4. One-year Kaplan-Meier survival curves by M-CDS classes (test set).**

population [17]. In particular, for all-cause deaths, the population attributable risk was estimated at 1.3% for schizophrenia [18] and 12.7% for depression [19]. Recent studies showed an increased mortality in patients with psychotic disorders after acute coronary syndrome [20] and breast cancer [21] compared with their counterparts without psychosis. Our findings corroborate the need to optimise the coordinated care of general medical conditions in those with mental disorders.

The predictive ability of M-CDS was significantly higher than that of the CCI, which is based on Hospital Discharge Records. A possible reason for the lower performance of the CCI is that Hospital Discharge Records may have high quality of primary diagnosis while the accuracy of secondary could be lower [22]. Moreover, HDR are often subject to restrictions on the number of diagnosis recorded, while our drug-based score does not have this limitation.

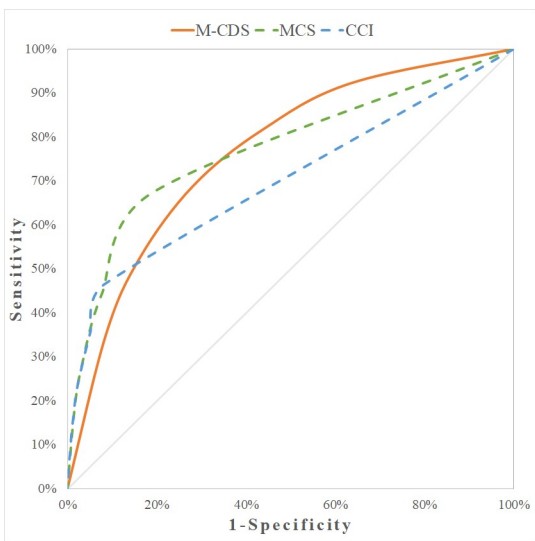

**Fig 5. ROC curves comparing M-CDS and MCS to predict 1-year mortality (test set).**

Our results are consistent with an Italian study, in which the Drug Derived Complexity Index (DDCI), based on prescription patterns indicative of 19 chronic diseases, proved to be superior to the CCI in predicting 1-year mortality in a large regional population [23].

Furthermore, M-CDS proved to have a discriminatory power similar to that of the MCS, although this latter is based on multiple data sources. Corrao et al. [11] found a difference between MCS and the old CDS of more than ten percentage AUC points, so, even though we have not directly tested the M-CDS against the CDS in our data, we assume that the M-CDS would have a better discriminatory power that the CDS.

The M-CDS weights are based on mortality, therefore when the score is used to predict 1-year hospitalizations it slightly loses its predictive power. Our intent was to develop a score useful for clinicians and researchers [24], while a recalibration of the weights of chronic conditions would be needed to predict healthcare resource utilization.

There are several advantages in using one-source morbidity score: data availability differs radically across countries (both high- or low-income) and it is not always possible to merge individual level data databases [25]; there is a computational simplification and a considerable time saving compared to scores that need multiple databases linkage and the loss in predictive performance is negligible; moreover, performance homogeneity is more easily ensured with the ATC code system, even for comparisons across geographical areas and over time, while diagnostic coding habits may differ.

The M-CDS has different potential applications. It may support policy makers, managers, clinicians in assessing the performance of health systems, population needs and in health policy planning, including resource allocations to local health districts or General Practitioners' remuneration. We are aware that other multisource and complex risk scores, e.g. Adjusted Clinical Groups (ACG), better capture the patient case-mix and explain variance in individual costs [26], however they are challenging to implement and expensive [27].

M-CDS can also be used for risk adjustment in real-world studies on treatment effectiveness and safety, where treatment arms are usually unbalanced for clinical characteristics.

Several potential limitations of this study should be acknowledged. First, data from dispensing databases do not indicate that the (full) package is consumed. Therefore, the actual exposure to the therapeutic regimen cannot be established. Second, the pharmaceutical prescription database only includes reimbursed dispensations, therefore private supply is not considered. However, this latter bias is partially mitigated by our focus on patients with the main clinically important chronic conditions, to whom drugs are provided free of charge by the National Health Service.

Moreover, the category "cardiovascular and cerebrovascular diseases" is very heterogeneous for possible impact on mortality, since it includes a variety of conditions, the most frequent of which is hypertension. Thus, the weight of each disease included in this category on 1-year mortality is not strictly reflected in the score.

We had no access to the Residents' Registry, so we were forced to use the OSD to select the population at baseline. Still, we were able to cover almost the entire population of interest (94.4%) by extracting all individuals who had at least one contact with outpatient services in the 18 months preceding the index year.

The external validity of score has been tested in a population which is very similar to the original population because it consists of residents in a bordering geographical area located in the same region. Our scoring system might perform differently in countries other than Italy due to possible small differences in both reimbursement policies and drug availability on the market. Its external validity in other health systems needs therefore to be tested. Eventually, periodic updates to the score should be done as new drugs or treatment recommendations are introduced.

Future perspectives include the application of other machine learning algorithms to investigate drug interactions and whether having multiple conditions increases the risk of death in an additive or multiplicative way.

## Conclusion

In conclusion, M-CDS index proved to be a good predictor of 1-year mortality. It reflects updated pharmaceutical prescriptions and has advantages compared to other indices because it is based on a single data source not affected by variability in diagnostic coding. Moreover, it encompasses a large number of conditions, therefore allowing in-depth studies on the interplay between mental and physical disorders.

The score is a simple and easy-to-implement instrument to stratify the population, to perform risk adjustment for case-mix in quality of care studies, potentially leading to an improvement in the management of chronic conditions, based on patients' needs and risks.

## Supporting information

**S1 Table. List of candidate conditions identified with their specific associated drugs and the minimum number of prescriptions (N).** General criteria: at least 2 prescriptions in the observed year. For drugs which unambiguously identify the specific diagnosis (e.g., drug listed for patients with multiple sclerosis) we considered only 1 prescription. For drugs frequent also in minor and non-chronic conditions (e.g., proton pump inhibitors for peptic ulcer) we considered at least 3 prescriptions. *nonspecific association; #wrong association; §not refunded drug.
(DOCX)

## Author Contributions

**Conceptualization:** Maria Pia Fantini, Elisabetta Poluzzi.

**Formal analysis:** Marica Iommi, Simona Rosa.

**Investigation:** Michele Fusaroli.

**Methodology:** Marica Iommi, Simona Rosa.

**Supervision:** Paola Rucci, Maria Pia Fantini, Elisabetta Poluzzi.

**Writing – original draft:** Marica Iommi, Simona Rosa, Michele Fusaroli.

**Writing – review & editing:** Marica Iommi, Simona Rosa, Michele Fusaroli, Paola Rucci, Maria Pia Fantini, Elisabetta Poluzzi.

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
