## [Decision Letter · Decision Letter 0]

27 Jul 2020

PONE-D-20-13521

Modified-Chronic Disease Score (M-CDS): predicting the individual risk of death using drug prescriptions

PLOS ONE

Dear Dr. Iommi,

Thank you for submitting your manuscript to PLOS ONE. After careful consideration, we feel that it has merit but does not fully meet PLOS ONE’s publication criteria as it currently stands. Therefore, we invite you to submit a revised version of the manuscript that addresses the points raised during the review process.

We look forward to receiving your revised manuscript.

Kind regards,

Kevin Lu, PhD

Academic Editor

PLOS ONE

2. In ethics statement in the manuscript and in the online submission form, please provide additional information about the patient records used in your retrospective study. Specifically, please ensure that you have discussed whether all data were fully anonymized before you accessed them and/or whether the IRB or ethics committee waived the requirement for informed consent. If patients provided informed written consent to have data from their medical records used in research, please include this information.

**Comments to the Author**

1. Is the manuscript technically sound, and do the data support the conclusions?

Reviewer #1: Yes

Reviewer #2: Yes

2. Has the statistical analysis been performed appropriately and rigorously? 

Reviewer #1: Yes

Reviewer #2: Yes

3. Have the authors made all data underlying the findings in their manuscript fully available?

Reviewer #1: Yes

Reviewer #2: Yes

4. Is the manuscript presented in an intelligible fashion and written in standard English?

Reviewer #1: Yes

Reviewer #2: Yes

5. Review Comments to the Author

Reviewer #1: This interesting study developed and validated the modified CDS score, which can be used to predict the one-year mortality among patients older than 50 years of age. The manuscript is well written and clearly presented. I have a few comments for the authors to consider.

1. The original CDS was developed in the general population. Also, it was developed to predict hospitalization and mortality. Please clarify why this modified-CDS was only developed in patients older than 50 years of age and why only mortality was evaluated.

2. I suggest the authors provide some details such as the penalty parameter of the LASSO procedure for variables selection. Without those details, it is not clear to me why and how the 18 variables were selected. More details of the CHAID analysis are also important for readers to understand how the categories were created.

3. The Imola population used in the external validation was very similar to the population used to develop the modified CDS. I would suggest more discussion about the external validity and generalizability of the modified CDS.

Reviewer #2: The authors proposed the modified Chronic Disease Score (M-CSD) using health administrative database in Northern Italy. The M-CDS was tested and compared on sample sets with different multimorbidity scores. Below are comments for authors to consider.

1. In the abstract section starting line 24, the format should be consistent with the format of abstract in page 1 where abstract section title “background”, “methods”, “results”, and “conclusions” were added.

2. In line 28, the authors mentioned that “the original version of the CDS has limitations”. However, the details of these limitations were not discussed throughout the paper. It would be helpful if the limitations of original CDS were explained in more detail in the introduction section.

3. In line 102, the meaning of abbreviation “OSC” is not clear.

4. In line 135-136, the authors wrote that “a cut-off on the minimum number of prescriptions was set for the years 2016 and 2017, depending on the drug”. What was the exact criteria of the minimum number of prescriptions for each drug?

5. In line 166-167, the authors claimed that “Of these, 1304 were excluded, because of missing data (age or gender)”. For patients who were excluded from 437865 individuals aged older than 50 years due to missing age, how were they included in the first step when one of the inclusion criteria is older than 50 years?

6. Please move all the tables and figure captions to the end of manuscript.

7. In the discussion section, could the authors discuss about the differences between CDS and M-CDS, and what are the advantages of M-CDS over CDS?

8. Another potential limitation is that the scoring system might work differently when it is used in a project with study period that is far away from 2016 and 2017.

9. The conclusion section is too concise. It almost has the same length as the conclusion in abstract. Please elaborate on the conclusion.

10. In line 293, the authors claimed that “The M-CDS proved to be a valid instrument to predict one-year mortality in the population aged 50 years or more”. However, the M-CDS was only observed, not proven, to perform better than CDS and similarly with MCS in test sets.

11. In line 295, the authors claimed that “The score can be easily used for risk adjust in real-world studies”. This claim should be limited to the real-world studies in Italy as the authors has already pointed out in the limitation that scoring system might perform differently in countries other than Italy.

6. PLOS authors have the option to publish the peer review history of their article (what does this mean?). If published, this will include your full peer review and any attached files.

Reviewer #1: **Yes: **Junjie Ma

Reviewer #2: No

---

## [Author Response · Author response to Decision Letter 0]

22 Sep 2020

We revised the style of figures and tables.

2. In ethics statement in the manuscript and in the online submission form, please provide additional information about the patient records used in your retrospective study. Specifically, please ensure that you have discussed whether all data were fully anonymized before you accessed them and/or whether the IRB or ethics committee waived the requirement for informed consent. If patients provided informed written consent to have data from their medical records used in research, please include this information.

We clarified the ethics statement in the methods section (line 95-104):

“The retrospective study was carried out in conformity with the regulations on data management with the Italian law on privacy (Legislation Decree 196/2003 amended by Legislation Decree 101/2018). Data were anonymized prior to the analysis at the regional statistical office, where each patient is assigned a unique identifier. This identifier does not allow others to trace the patient’s identity and other sensitive data. Anonymized regional administrative data can be used without a specific written informed consent when patient information is collected for healthcare management and healthcare quality evaluation and improvement (according to art. 110 on medical and biomedical and epidemiological research, Legislation Decree 101/2018). The study was approved by the Independent ethics committee of the Larger Area Emilia Center (CE-AVEC) of the Emilia-Romagna Region, study protocol 398/2019/Oss/AOUBo on 19/06/2019.”

The datasets generated and/or analysed during the current study are property of a third party that is Emilia-Romagna Regional Health Agency (https://assr.regione.emilia-romagna.it/) and, although they are anonymized, datasets are not publicly available due to the current regulation on privacy. 

The description of the administrative databases is available from the website https://salute.regione.emilia-romagna.it/siseps/sanita/asa/documentazione.

The database includes the following variables: patient ID, age, gender, area of residence, 33 dummy variables for drug prescriptions, death at one year (yes/no) and days of follow-up.

Reviewer #1: 

This interesting study developed and validated the modified CDS score, which can be used to predict the one-year mortality among patients older than 50 years of age. The manuscript is well written and clearly presented. I have a few comments for the authors to consider.

1. The original CDS was developed in the general population. Also, it was developed to predict hospitalization and mortality. Please clarify why this modified-CDS was only developed in patients older than 50 years of age and why only mortality was evaluated.

Our purpose was to develop an accurate index to predict one-year mortality among adults aged ≥50 years, who are at risk of having at least one chronic condition, for clinical and research use. 

As to the ability of the MCS score to predict hospitalizations, we have now added further analyses in the methods section (line 179-180):

“As a secondary outcome, the ability of M-CDS to predict 1-year hospitalization was further evaluated.”

In the results section (line 251-257):

“Secondary outcome

We also evaluated 1-year hospitalization as secondary outcome. In the Bologna training set the AUC was 0.710 (95% CI [0.660-0.760]), slightly higher than the test set (AUC 0.695; 95% CI [0.594-0.795]) and the Imola population (AUC 0.667; 95% CI [0.570-0.764]).

The M-CDS, evaluated in the test set, did not significantly differ from either the CCI (AUC 0.659; 95% CI [0.517-0.801]; χ2 test=0.21; p=0.647) or the MCS (AUC 0.786; 95% CI [0.673-0.899]; χ2 test=2.14; p=0.143).”

And in the discussion section (line 295-298):

“The M-CDS weights are based on mortality, therefore when the score is used to predict 1-year hospitalizations it slightly loses its predictive power. Our intent was to develop a score useful for clinicians and researchers (23), while a recalibration of the weights of chronic conditions would be needed to predict healthcare resource utilization.”

2. I suggest the authors provide some details such as the penalty parameter of the LASSO procedure for variables selection. Without those details, it is not clear to me why and how the 18 variables were selected. More details of the CHAID analysis are also important for readers to understand how the categories were created.

For the LASSO procedure we chose an α=1 and for variable selection we used a λ value within one standard error from the minimum (in this case λ= 0.0014). 

We added to the methods section more details about the LASSO procedure and of the classification tree analysis (line 164-165 and line 170-171). 

“To reduce the risk of overfitting, we selected the most parsimonious model, that is within one standard error of the best model (13).”

“Finally, a classification tree analysis with chi-square automatic interaction detection (CHAID) growing process (maximum tree depth=3; minimum cases in parent node=1000, child node=500; significance level of spitting and merge set to 0.05) was used to identify optimal cut-point of the total M-CDS to predict one-year mortality (15). ”

3. The Imola population used in the external validation was very similar to the population used to develop the modified CDS. I would suggest more discussion about the external validity and generalizability of the modified CDS.

We added this point as a potential limitation (line 327-328):

“The external validity of score has been tested in a population which is very similar to the original population, because it consists of residents in a bordering geographical area located in the same region.”

 

Reviewer #2: 

The authors proposed the modified Chronic Disease Score (M-CSD) using health administrative database in Northern Italy. The M-CDS was tested and compared on sample sets with different multimorbidity scores. Below are comments for authors to consider.

1. In the abstract section starting line 24, the format should be consistent with the format of abstract in page 1 where abstract section title “background”, “methods”, “results”, and “conclusions” were added.

Done (line 24, 31, 40 and 47).

2. In line 28, the authors mentioned that “the original version of the CDS has limitations”. However, the details of these limitations were not discussed throughout the paper. It would be helpful if the limitations of original CDS were explained in more detail in the introduction section.

We added some details on limitations of CDS in the Introduction (line 70-77). Maximum number of words in the abstract does not allow to include additional paragraphs. 

We add these paragraphs in the introduction section:

“No further updates have been made, inevitably leading to limitations in the use of the score, as new drugs have been introduced in the market to treat chronic diseases (e.g., monoclonal antibodies for autoimmune disorders). Furthermore, in drug prescription databases specific criteria (e.g., amount threshold) to select chronic patients are required.

Several studies have compared the predictive ability of diagnosis-based indexes to drug-based ones, however, the superiority of an approach in predicting health status has not been demonstrated yet (8,9), and the lack of an updated version of CDS does not allow appropriate performance assessment and comparisons (10).”

3. In line 102, the meaning of abbreviation “OSC” is not clear.

We apologize for the typo. It is the abbreviation for Outpatient Speciality Database (OSD) (line 120).

“To compute the Charlson Comorbidity Index and the Multisource Comorbidity Score, the Hospital Discharge Record (HDR) database of 2016-2017 were also linked with the OSD”

4. In line 135-136, the authors wrote that “a cut-off on the minimum number of prescriptions was set for the years 2016 and 2017, depending on the drug”. What was the exact criteria of the minimum number of prescriptions for each drug?

Thresholds are defined based on use recommendation, marketed packages, and prescribing/reimbursing rules. Details have been provided in the text (line 155-157):

“The list of candidate condition, the corresponding ATC codes and the minimum number of prescriptions are reported in Table S1. Thresholds of number of prescriptions are defined on the basis of recommended dosing schemes, packages available on the market and relevant prescribing/reimbursing rules.”

And in the description of the Table S1 in the supporting materials:

“S1 Table. List of candidate conditions identified with their specific associated drugs and the minimum number of prescriptions (N). General criteria: at least 2 prescriptions in the observed year. For drugs which unambiguously identify the specific diagnosis (e.g., drug listed for patients with multiple sclerosis) we considered only 1 prescription. For drugs frequent also in minor and non-chronic conditions (e.g., proton pump inhibitors for peptic ulcer) we considered at least 3 prescriptions.”

5. In line 166-167, the authors claimed that “Of these, 1304 were excluded, because of missing data (age or gender)”. For patients who were excluded from 437865 individuals aged older than 50 years due to missing age, how were they included in the first step when one of the inclusion criteria is older than 50 years?

We extracted all the residents of the LHA of Bologna and Imola, then we excluded individual with missing age or gender and then we selected people aged ≥50 years (line 188-189).

“We identified 436,561 individuals aged ≥ 50 years, resident in the LHAs of Bologna and Imola, alive on January 1st, 2018, with at least 1 outpatient service access in the previous 18 months. Of these, 1,304 were excluded, because of missing data (age or gender).”

6. Please move all the tables and figure captions to the end of manuscript.

We copied the captions of all the tables and figures to the end of manuscript. We also kept each figure caption directly after the paragraph as requested by the journal.

7. In the discussion section, could the authors discuss about the differences between CDS and M-CDS, and what are the advantages of M-CDS over CDS?

Compared to previous versions of CDS, M-CDS represents an unambiguous method of application of the score to prescribing data: we updated each chronic condition and we identified new and relevant ATC codes, establishing the minimum number of prescriptions to define each specific condition, and we selected only conditions actually impacting on mortality. 

The performance of the old CDS was not assessed since its application anyway needs some specific subjective steps (ATC code assignment, threshold definition…). As a matter of fact, many previous studies using CDS are lacking specific steps of application... (e.g., Corrao et al. 2017).

We added this sentence to the discussion section (line 291-294):

“Corrao et al. (11) found a difference between MCS and the old CDS of more than ten percentage AUC points , so, even though we have not directly tested the M-CDS against the CDS in our date, we assume that the M-CDS would have a better discriminatory power that the CDS.”

8. Another potential limitation is that the scoring system might work differently when it is used in a project with study period that is far away from 2016 and 2017.

We thank the reviewer for this important comment. The study population consists of people resident in the study areas in 2018, which is not too far away from 2020. We are aware that the scoring system might perform differently if the epidemiology of the population changes, for instance as a result of the recent COVID pandemic. Therefore, the performance of the score should be tested over time. 

We have added a sentence to the manuscript (line 330-332):

“Its external validity in other health systems needs therefore to be tested. Eventually, periodic updates to the score should be done as new drugs or treatment recommendations are introduced.”

9. The conclusion section is too concise. It almost has the same length as the conclusion in abstract. Please elaborate on the conclusion.

We have now expanded the conclusion section, as suggested. 

10. In line 293, the authors claimed that “The M-CDS proved to be a valid instrument to predict one-year mortality in the population aged 50 years or more”. However, the M-CDS was only observed, not proven, to perform better than CDS and similarly with MCS in test sets.

Please see response to point 7. 

11. In line 295, the authors claimed that “The score can be easily used for risk adjust in real-world studies”. This claim should be limited to the real-world studies in Italy as the authors has already pointed out in the limitation that scoring system might perform differently in countries other than Italy.

We revised the paragraph in the discussion section (line 339-346):

“In conclusion, M-CDS index proved to be a good predictor of 1-year mortality. It reflects updated pharmaceutical prescriptions and has advantages compared to other indices because it is based on a single data source not affected by variability in diagnostic coding. Moreover, it encompasses a large number of conditions, therefore allowing in-depth studies on the interplay between mental and physical disorders. 

The score is a simple and easy-to-implement instrument to stratify the population, to perform risk adjustment for case-mix in quality of care studies, potentially leading to an improvement in the management of chronic conditions, based on patients’ needs and risks.”

---

## [Decision Letter · Decision Letter 1]

6 Oct 2020

Modified-Chronic Disease Score (M-CDS): predicting the individual risk of death using drug prescriptions

PONE-D-20-13521R1

Dear Dr. Iommi,

We’re pleased to inform you that your manuscript has been judged scientifically suitable for publication and will be formally accepted for publication once it meets all outstanding technical requirements.

Kind regards,

Kevin Lu, PhD

Academic Editor

PLOS ONE

Reviewers' comments:

Reviewer's Responses to Questions

**Comments to the Author**

1. If the authors have adequately addressed your comments raised in a previous round of review and you feel that this manuscript is now acceptable for publication, you may indicate that here to bypass the “Comments to the Author” section, enter your conflict of interest statement in the “Confidential to Editor” section, and submit your "Accept" recommendation.

Reviewer #1: All comments have been addressed

Reviewer #2: All comments have been addressed

2. Is the manuscript technically sound, and do the data support the conclusions?

Reviewer #1: Yes

Reviewer #2: Yes

3. Has the statistical analysis been performed appropriately and rigorously? 

Reviewer #1: Yes

Reviewer #2: Yes

4. Have the authors made all data underlying the findings in their manuscript fully available?

Reviewer #1: Yes

Reviewer #2: Yes

5. Is the manuscript presented in an intelligible fashion and written in standard English?

Reviewer #1: Yes

Reviewer #2: Yes

6. Review Comments to the Author

Reviewer #1: (No Response)

Reviewer #2: (No Response)

7. PLOS authors have the option to publish the peer review history of their article (what does this mean?). If published, this will include your full peer review and any attached files.

Reviewer #1: **Yes: **Junjie Ma

Reviewer #2: No

---

## [Editor Report · Acceptance letter]

8 Oct 2020

PONE-D-20-13521R1 

Modified-Chronic Disease Score (M-CDS): predicting the individual risk of death using drug prescriptions 

Dear Dr. Iommi:

I'm pleased to inform you that your manuscript has been deemed suitable for publication in PLOS ONE. Congratulations! Your manuscript is now with our production department. 

Kind regards, 

on behalf of

Professor Kevin Lu 

Academic Editor

PLOS ONE